# ConFu: Contemplate the Future for Better Speculative Sampling

**Zongyue Qin** [* 1]  **Raghavv Goel** [* 2]  **Mukul Gagrani** [2]  **Risheek Garrepalli** [2]  **Mingu Lee** [γ 2]  **Yizhou Sun** [γ 1]

## Abstract

Speculative decoding has emerged as a powerful approach to accelerate large language model (LLM) inference by employing lightweight draft models to propose candidate tokens that are subsequently verified by the target model. The effectiveness of this paradigm critically depends on the quality of the draft model. While recent advances such as the EAGLE series achieve state-of-the-art speedup, existing draft models remain limited by error accumulation: they condition only on the current prefix, causing their predictions to drift from the target model over steps. In this work, we propose **ConFu** (Contemplate the Future), a novel speculative decoding framework that enables draft models to anticipate the future direction of generation. ConFu introduces (i) *contemplate tokens* and *soft prompts* that allow the draft model to leverage future-oriented signals from the target model at negligible cost, (ii) a *dynamic contemplate token mechanism with MoE* to enable context-aware future prediction, and (iii) a training framework with *anchor token sampling* and *future prediction replication* that learns robust future prediction. ConFu improves token acceptance rates and generation speed over EAGLE-3 by 8–11% on Llama-3 3B/8B and by approximately 20% on Qwen-3 4B across downstream tasks. We believe our work is the first to bridge speculative decoding with continuous reasoning tokens, offering a new direction for accelerating LLM inference.

## 1. Introduction

Large language models (LLMs) have achieved remarkable performance across a wide range of natural language pro-

cessing tasks, yet their inference remains prohibitively expensive due to the autoregressive nature of text generation. Each decoding step requires a forward pass through the full model, resulting in high latency and computational cost. To mitigate this issue, a growing body of work has explored *speculative decoding* (Leviathan et al., 2023; Miao et al., 2024; Qin et al., 2025b;a; Li et al., 2024a;b; 2025; Goel et al., 2024; Jeon et al., 2024), an inference paradigm that employs a lightweight *draft model* to propose candidate tokens which are subsequently verified by the target model. By amortizing multiple draft tokens within a single verification pass of the target model, speculative decoding can accelerate generation without compromising the quality of outputs.

A central factor determining the effectiveness of speculative decoding is the quality of the draft model. Recent advances have led to a series of draft models with increasingly strong predictive capabilities. Notably, the **EAGLE** family (Li et al., 2024a;b; 2025) represents the state of the art in speculative decoding. EAGLE-1 (Li et al., 2024a) first demonstrated the effectiveness of training a single-layer transformer that exploits the hidden states of the target model to generate draft tokens autoregressively. EAGLE-2 (Li et al., 2024b) introduced a new technique of context-aware dynamic draft tree into drafting modeling. EAGLE-3 further enhanced both architecture and training framework, setting new benchmarks in speculative decoding speed. Across diverse benchmarks, the EAGLE models consistently deliver superior speedups compared to prior draft models (Cai et al., 2024; Zhang et al., 2025), and are recognized as the current best-in-class approach.

Despite these successes, existing draft models, including the EAGLE series, have a shared drawback: they generate draft tokens by conditioning solely on the current prefix. This design is prone to error accumulation. As shown in Figure 1a, at first the hidden representations of the draft model align well with those of the target model, yielding accurate predictions. However, as the decoding proceeds, small errors accumulate, the draft distribution drifts from the target distribution, and token acceptance rates decline. This misalignment undermines the potential efficiency gains of speculative decoding.

In this work, we argue that draft models should not merely

---

[*]Equal contribution  [1]University of California Los Angeles, United States [2]Qualcomm AI Research, United States. Correspondence to: Zongyue Qin <qinzongyue@cs.ucla.edu>, Raghavv Goel, Mukul Gagrani, Mingu Lee <raghgoel,mgagrani,mingul@qti.qualcomm.com>.

*Proceedings of the $43^{rd}$ International Conference on Machine Learning*, Seoul, South Korea. PMLR 306, 2026. Copyright 2026 by the author(s).

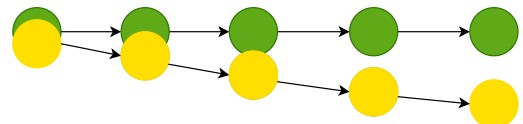

*(a)* Draft model hidden representations without future prediction.

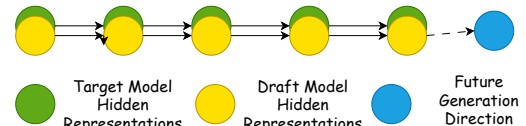

*(b)* Draft model hidden representations with future prediction.

*Figure 1.* Illustration of the purpose of future generation direction prediction

focus on predicting the immediate next token, but should also anticipate the *future direction* of generation. Intuitively, before committing to specific token choices, a draft model can benefit from understanding what the target model is planning to generate next at a higher level, namely, the target model's current "thought". As illustrated in Figure 1b, if the draft model is provided with information about the target model's current "thought" and is encouraged to draft tokens that follow this direction, it becomes more likely to propose candidates that stay on the same semantic trajectory as planned by the target model. As a result, the draft tokens are more accurate, and therefore less likely to be rejected during the verification stage.

We instantiate this idea in ConFu (Contemplate the Future), a novel speculative decoding framework. ConFu introduces three key innovations. First, we introduce *contemplate tokens* and *soft prompts* that encourage the target model to expose signals of its intermediate reasoning with minimal additional inference cost. These signals are then provided to the draft model as auxiliary inputs, enabling more accurate and reliable token drafting. Second, we propose a *dynamic contemplate token mechanism based on Mixture-of-Experts (MoE)*, which allows contemplate tokens to adapt to diverse contexts and achieve greater expressive capacity. Third, we develop a training framework based on *anchor token sampling* and *future prediction replication*, which efficiently and effectively trains the model to learn robust future predictions.

Experiments on SpecBench (Xia et al., 2024) demonstrate that ConFu consistently improves both token acceptance rates and decoding speed over the state-of-the-art speculative decoding baseline, EAGLE-3 (Li et al., 2025). Across a wide range of downstream tasks, including writing, question answering, summarization, translation, coding, and mathematical reasoning, ConFu achieves substantial gains under diverse decoding conditions. On average, ConFu improves token acceptance rates and generation speed by 8-11% with

Llama-3 3B and 8B models. These improvements are consistent across all task categories, sampling temperatures, and computation budgets.

More broadly, our results suggest that speculative decoding can be significantly strengthened by equipping draft models with the ability to *contemplate the future*. By conditioning draft generation on the target model's predicted semantic trajectory, ConFu produces draft tokens that align more closely with the target distribution, thereby reducing rejection rates during verification and improving overall throughput. At high-level glance, EAGLE (Li et al., 2024a) introduced a method for adding target-biased guidance to draft model and subsequent works have been to mitigate training and inference mismatch (Li et al., 2025; Zhang et al., 2025; Hu et al., 2025). In this work, we provide a new direction for improving draft generation by additionally conditioning the draft model with contemplate token and future token. We view ConFu as an important step toward integrating speculative decoding with latent reasoning paradigms (Hao et al., 2025; Cheng & Van Durme, 2024; Shen et al., 2025). To the best of our knowledge, this is the first work to explicitly bridge speculative decoding with continuous latent "thought" representations, opening a new direction for accelerating LLM inference through future-aware generation.

## 2. Preliminaries

**Speculative decoding** utilizes a small, fast *draft model* ($M_d$) to generate a sequence of candidate tokens, which are then verified in a single, parallel forward pass by the large, powerful *target model* ($M_t$) (Leviathan et al., 2023; Miao et al., 2024).

In its standard form, the process works as follows:

1. **Drafting:** Given a prompt or a previously generated sequence $x_{1:n}$, the draft model $M_d$ autoregressively generates a short sequence of $K$ draft tokens, $\tilde{x}_{n+1}, \ldots, \tilde{x}_{n+K}$.

2. **Verification:** The target model $M_t$ takes the combined sequence $x_{1:n}, \tilde{x}_{n+1}, \ldots, \tilde{x}_{n+K}$ as input and performs a single forward pass to compute the probability distributions for the next token at each position.

3. **Acceptance/Rejection:** The draft tokens are checked sequentially. For each position $i$ from 1 to $K$, the draft token $\tilde{x}_{n+i}$ is accepted if it matches the token sampled from the target model's distribution $p_t(\cdot|x_{1:n}, \tilde{x}_{n+1}, \ldots, \tilde{x}_{n+i-1})$. If a token is accepted, the process continues to the next one. If a token is rejected, it and all subsequent draft tokens are discarded.

4. **Correction:** The first token that was rejected is replaced by a new token sampled from the target model's

corrected distribution at that position. The final accepted sequence becomes the input for the next drafting step.

The speedup comes from the number of tokens accepted in a single verification step, effectively replacing multiple sequential forward passes of the target model with one.

To improve the acceptance rate, the drafting process can be extended to generate a **tree of candidate tokens** instead of a single linear sequence (Miao et al., 2024; Sun et al., 2023). The draft model proposes multiple potential tokens at each step, creating a tree of draft tokens. The target model then validates all paths in this tree in parallel using a tree attention mechanism. The longest path that is consistent with the target model's predictions is accepted. This approach increases the likelihood that at least one drafted sequence will be correct, leading to a higher average number of accepted tokens per step.

**EAGLE** (Li et al., 2024a;b; 2025) is an advanced speculative decoding framework that addresses the core challenge of low acceptance rates by eliminating the need for a separate, misaligned draft model. Instead, it integrates the drafting mechanism directly into the target model itself.

The key innovation in EAGLE is the use of lightweight draft heads. The draft model can be seen as a single-layer transformer model that exploits the hidden states of the target model. By exploiting the target model's hidden representations, the EAGLE draft model achieves high acceptance rate for the draft tokens. And due to its lightweight architecture, the cost of generating draft tokens is much smaller than running an independent draft model. EAGLE-3 further improves the architecture of EAGLE by utilizing the hidden states of the target model from multiple layers. Specifically, EAGLE-3 concatenates target hidden-states from initial, middle, and final layer as $\boldsymbol{h}_t^{M_t,cat} \in \mathbb{R}^{3d}$ which is then down-projected to obtain, $h_t^{M_d} = \boldsymbol{W}_{proj}\boldsymbol{h}_t^{M_t,cat} \in \mathbb{R}^d$. The draft model then utilizes the hidden state $h_t^{M_d}$ to generate draft tokens autoregressively.

## 3. ConFu: The Methodology

In this section, we introduce our model architecture design and how the draft model is trained. Specifically, Section 3.1 introduces the overall architecture of ConFu and the inference framework with contemplate tokens. Then Section 3.2 illustrates how we utilize MoE to achieve dynamic contemplate tokens. Finally, Section 3.3 illustrates how ConFu is trained.

### 3.1. Capture Future with Contemplate Tokens

The goal of future prediction is to generate a continuous embedding that captures the current "thought" of the target

model, which can then guide the draft model in sampling more accurate future tokens. Two key requirements must be satisfied: (1) the future prediction module must have *sufficient capacity* to approximate the target model's internal reasoning, and (2) it should incur *minimal additional cost* during inference.

Recent studies on latent reasoning demonstrate that LLMs can generate continuous "thought tokens", after post-training, which serve as intermediate reasoning states (Hao et al., 2025; Cheng & Van Durme, 2024; Shen et al., 2025). While effective, generating such tokens requires an autoregressive process with multiple forward passes of the target model, which is prohibitively expensive. Instead, we propose to exploit *contemplate tokens*, also known as *pause tokens*[1] (Goyal et al., 2024). A pause token is a special token appended to the input prefix that causes the LLM to perform additional computation before producing the next output. Goyal et al. (2024) observed that introducing pause tokens improves reasoning accuracy, and attributed this effect to the fact that adding pause tokens can be viewed as increasing the hidden representations of the model when compute the next token. From another perspective, the hidden representations of pause tokens encode the model's intermediate "thoughts". More importantly, pause tokens can be processed in parallel with other input tokens, resulting in negligible extra inference cost. This makes them a promising mechanism for future prediction.

A challenge, however, is that speculative decoding does not permit fine-tuning the target model, as doing so would alter model behavior. Simply learning an embedding for the contemplate token may also be insufficient to capture meaningful future predictions. To address this, we draw inspiration from BiTA (Lin et al., 2025) and utilize learnable *soft prompt tokens* as auxiliary parameters that instruct the target model to produce future prediction.

As illustrated in Figure 2, we prepend a set of prompt tokens to the target model's KV cache and append a contemplate token to the current input prefix. Formally, the prompt tokens are *learnable embeddings* with the same dimensionality as the target model's KV cache. The contemplate tokens can similarly be implemented as learnable token embeddings, following prior work on pause tokens (Goyal et al., 2024; Lin et al., 2025). In ConFu, however, we further extend contemplate tokens beyond static embeddings by allowing them to become *dynamic* during inference. We will describe this mechanism in the next subsection.

During training, the target model is frozen, while both the soft prompt tokens and the contemplate token embedding are optimized. Notably, the attention mask is modified such that only contemplate tokens can attend to the soft prompt

---

[1]In this paper, we use these two terms interchangeably.

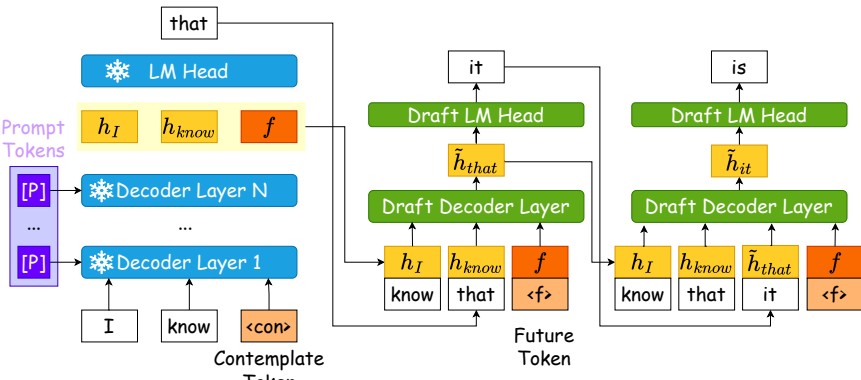

*Figure 2.* Overview of ConFu's inference pipeline. Given the input tokens, the target model first produces the next output token along with a future prediction vector $f$, using both prompt tokens and contemplate tokens. The draft model then conditions on $f$ as an additional future token to autoregressively generate draft tokens. Throughout the drafting process, the future token $f$ remains fixed and is always appended to the end of the input sequence.

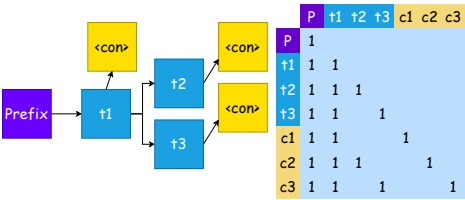

*Figure 3.* Verification with contemplate tokens in ConFu. Let $t_1, t_2, t_3$ denote draft tokens in the speculative tree. We insert one contemplate token after each draft token so that the target model can simultaneously verify draft candidates and generate the corresponding future predictions. The tree attention mask is adjusted accordingly to ensure correct verification and alignment of future predictions with accepted tokens.

tokens, ensuring that the input prefix representations remain unaffected.

**Inference with Contemplate Tokens**  Figure 2 summarizes the overall inference procedure of ConFu. Unlike BiTA, which directly decodes future tokens from the hidden representations of contemplate tokens, ConFu instead uses these representations to guide draft generation. Specifically, the hidden state of the contemplate token is provided as an additional token to a lightweight draft model (implemented as a single-layer Transformer, similar to EAGLE). Conditioned on a shared $f$, the draft model can generate multiple steps of candidate tokens that better anticipate the target model's future trajectory, thereby improving the effectiveness of speculative sampling.

For the draft model, incorporating future information is lightweight: it only requires appending a single auxiliary token $f$, which can be processed efficiently alongside the existing tokens. However, for the target model, each speculative iteration must perform two tasks simultaneously: (i)

verify the proposed draft tokens, and (ii) produce the future prediction for the next iteration.

A key challenge is that the future prediction must correspond to the final accepted draft token, which is not known in advance. To address this, we augment the draft token tree with $T$ contemplate tokens, inserting one contemplate token for each draft node and modifying the tree attention accordingly, as illustrated in Figure 3. This allows the target model to generate a distinct future prediction for every draft candidate in parallel. After verification, the future prediction associated with the last accepted token is selected and passed to the draft model in the next iteration.

We emphasize that the additional overhead introduced by contemplate tokens is modest. Let $t$ denote the prefix length, $s$ the number of soft prompt tokens (typically small, e.g., $s = 16$), and $T$ the number of draft nodes in the speculative tree. During the first iteration (target-model prefill), only a single contemplate token is appended, yielding a context length of $t + s + 1$. In later iterations, the target model verifies the speculative draft tree of size $T$. Because each draft node is paired with an inserted contemplate token, the target model processes a total of $2T$ tokens in parallel during verification. Since $T$ is typically moderate (e.g., $T = 30$), the resulting increase in computation remains small compared to the overall cost of target-model decoding.

### 3.2. Dynamic Contemplate Tokens with MoE

The soft prompt tokens and the contemplate token can be interpreted as specialized instructions that prompt the target model to summarize its current thought. However, due to the diversity of contexts encountered during generation, a single fixed instruction is often insufficient to elicit an accurate and faithful summarization. For instance, in mathematical reasoning, an instruction such as "my next equation is:" may

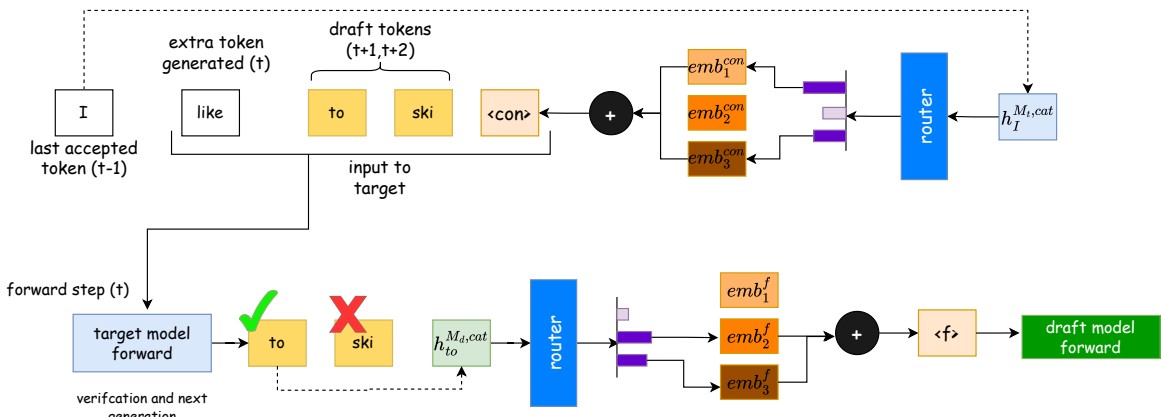

*Figure 4.* Illustration of Dynamic Contemplate Tokens with MoE. The input tokens contain both accepted tokens and the draft tokens of the current iteration. The MoE module only takes the hidden representation of *the last accepted token* as inputs. Then it computes the expert weights with a linear layer (router) and outputs the weighted sum of the selected learnable embeddings as the final contemplate token embedding. For simplicity a single [con] token is shown instead of 3 (1 for 'like' and 2 for draft tokens)

be more appropriate, whereas in long-form writing tasks, an instruction like "this paragraph is about:" can better capture the underlying intent. Therefore, a fixed contemplate token embedding might not be sufficient to capture the thought of the target model accurately across diverse tasks.

To address this limitation, we depart from prior work (Goyal et al., 2024; Lin et al., 2025), which models the contemplate token as a single learnable embedding. Instead, we parameterize the contemplate token using a Mixture-of-Experts (MoE) architecture, conditioned on the hidden state of the most recently accepted token.

Specifically, both the contemplate token embedding [con] (fed as input to the target model) and the future token embedding [f] (fed as input to the draft model) in Figure 2 are produced by two separate Mixture-of-Experts (MoE) modules. The [con] token is processed during draft token verification by target model, and therefore uses concatenated hidden-state of last accepted token ($h^{M_t,cat}$ defined in Section 2). The [f] is processed during draft model generation and uses the latest accepted draft token's hidden state in the draft model ($h^{M_d}$ in Section 2). Figure 4 shows the exact MoE modules for both [con] and [f]. The embedding MoE maintains a set of $n_{\text{expert}}$ learnable token embeddings, which serve as the experts. During inference, the MoE module takes as input the hidden state of the most recently accepted (or generated) token. A linear layer maps this hidden state to a set of logits over the experts, which are then normalized using a Softmax function. The top-$K_{\text{expert}}$ experts are selected, and the final token embedding is computed as a weighted linear combination of their embeddings, where the weights are given by the normalized gating scores.

This design allows the contemplate token to adaptively select among multiple expert instructions based on the current

context, enabling more accurate and context-aware future direction prediction. We believe this is the first instance of enabling dynamic-ness in the pause token setup.

### 3.3. Training Pipeline

Our draft model head is architecturally similar to the drafting head in EAGLE-3 (Li et al., 2025), with the key difference that we incorporate future prediction as an additional token. As a result, we adopt the same training objective as prior work. Given input tokens $x_{1:N}$, the draft model is trained to predict the next $L$ tokens under the train-time testing framework (Zhang et al., 2025; Li et al., 2025). Formally, the loss is defined as

$$
\sum_{t=1}^{N} \sum_{i=1}^{L} \text{KL} \Big[ P_{M_t}(x_{t+i} \mid x_{1:t+i-1}),
$$
$$
P_{M_d}(x_{t+i} \mid x_{1:t+i-1}, \boldsymbol{h}_{1:t}^{M_d}, \tilde{\boldsymbol{h}}_{t+1:t+i-1}) \Big] \tag{1}
$$

where $KL$ is the KL-divergence, $P_{M_t}$ and $P_{M_d}$ denote the output distributions of the target and draft models, respectively; $x_{1:t+L}$ is the training sequence; $\boldsymbol{h}_{1:t}^{M_d}$ are the down-projection of the target model's concatenated hidden representations used by draft model for $x_{1:t}$ as mentioned in Section 2; and $\tilde{\boldsymbol{h}}_{t+1:t+i-1}$ are the draft model's hidden representations for $x_{t+1:t+i-1}$.

**Efficient Training with Anchor Token Sampling** During training, a contemplate token must be inserted for each token position, which would double the sequence length and substantially increase memory consumption. To mitigate this issue, we adopt a memory-efficient training strategy based on *anchor token sampling*. Specifically, from a training sequence $x_{1:N}$, we randomly sample $K_{\text{train}}$ tokens as

a set of *anchor tokens* $T_{\text{anchor}}$. We only insert contemplate tokens for anchor tokens, and compute the loss over the next $L$ tokens following each anchor token. The resulting loss is

$$\sum_{t \in T_{\text{anchor}}} \sum_{i=1}^{L} \text{KL}\Big[P_{M_t}(x_{t+i} \mid x_{1:t+i-1}),$$
$$P_{M_d}(x_{t+i} \mid x_{1:t+i-1}, [\text{f}]_t, \boldsymbol{h}_{1:t}^{M_d}, \tilde{\boldsymbol{h}}_{t+1:t+i-1}, \boldsymbol{f}_t)\Big] \quad (2)$$

where, $f_t$ is last layer-hidden state of contemplate token at position $t$ conditioned on target input and prompt tokens. With this strategy, the sequence length increases from $N$ to $N + K_{\text{train}}$, instead of $2N$, substantially reducing memory overhead.

**Robust Training with Future Prediction Replication** Intuitively, since the future prediction $\boldsymbol{f}$ captures high-level intent or latent reasoning of the target model, it should be robust to small positional perturbations. That is, nearby tokens are expected to share similar future predictions. To encourage this robustness, we introduce a robust training strategy.

Let $\boldsymbol{f}_t$ denote the future prediction associated with an anchor token $x_t$. For a window of nearby tokens $\{x_{t+j}\}_{j=1}^{l}$ that are not selected as anchor tokens, where $l$ is a hyperparameter, we reuse $\boldsymbol{f}_t$ as their future prediction. The draft model is then trained to predict the next $L$ tokens for each $x_{t+j}$ using the same future prediction. The resulting training objective is

$$\sum_{t \in T_{\text{anchor}}} \sum_{j=0}^{l} \sum_{i=1}^{L} \text{KL}\Big[P_{M_t}(x_{t+j+i} \mid x_{1:t+j+i-1}),$$
$$P_{M_d}(x_{t+j+i} \mid x_{1:t+j+i-1}, [\text{f}]_t,$$
$$\boldsymbol{h}_{1:t+j}^{M_d}, \tilde{\boldsymbol{h}}_{t+j+1:t+j+i-1}, \boldsymbol{f}_t)\Big] \quad (3)$$

This loss implicitly encourages the soft prompt tokens and contemplate tokens to produce informative and robust future predictions that improve draft accuracy. Thus, no additional auxiliary losses are required to train the draft model.

## 4. Experiment

In this section, we evaluate the performance of ConFu and compare it against EAGLE-3 (Li et al., 2025), a state-of-the-art draft model that consistently outperforms prior speculative decoding approaches such as Medusa (Cai et al., 2024) and HASS (Zhang et al., 2025). We conduct experiments with Llama-3.2-3B-Instruct and Llama-3.1-8B-Instruct (Grattafiori et al., 2024) as target models.

**Training Setup** For the 8B target model, we use the official EAGLE-3 draft checkpoint[2]. For the 3B target model, since

---

[2] https://huggingface.co/yuhuili/

no public EAGLE-3 checkpoint is available, we train it from scratch using the official implementation[3]. Following Li et al. (2025), we train on the ShareGPT and UltraChat-200K (Ding et al., 2023) instruction datasets.

Since ConFu builds directly on the EAGLE-3 draft architecture, to save training time, we initialize ConFu from the corresponding trained EAGLE-3 checkpoints and further train it on the same data under identical optimization settings. We also experimented with continuing to train the EAGLE-3 baseline for the same number of additional steps, but observed no measurable improvement, ensuring that our gains are not due to longer training. Overall, this setup provides a fair and controlled comparison between ConFu and EAGLE-3. All the training are conducted with 8 NVIDIA-H100 GPUs.

**Evaluation Setup** We evaluate ConFu and EAGLE-3 on SpecBench (Xia et al., 2024), a comprehensive benchmark designed to assess speculative decoding performance across diverse instruction-following tasks, including writing, question answering, summarization, translation, coding, mathematical reasoning and other tasks.

All experiments are conducted on a single NVIDIA H100 GPU with batch size 1. For each method, we report two standard efficiency metrics: (i) the average accepted draft length ($\tau$), which measures how many draft tokens are accepted per verification step and the bonus token, and (ii) the speed-up ratio (SR) relative to standard autoregressive decoding, which captures end-to-end decoding speed improvements. Unless otherwise specified, both methods use the same decoding configurations (e.g., draft budget, sampling temperature) to ensure comparability.

### 4.1. Main Results

The comparison results are reported in Tables 1 and 2. We vary the sampling temperature ($T \in 0, 0.7, 1.0$) and the number of draft nodes ($30, 60$). Across both target models, ConFu consistently outperforms EAGLE-3 under all evaluated decoding configurations, achieving higher accept length and speed-up ratio (SR).

**Effect of Temperature.** The advantage of ConFu is most pronounced at lower sampling temperatures. For example, under greedy decoding ($T = 0$) with 30 draft nodes, ConFu improves the speed-up ratio by approximately $1.14\times$ and $1.15\times$, and increases accept length by $9.2\%$ and $12.8\%$ for the 8B and 3B target models, respectively. We attribute this trend to the fact that lower temperatures induce a sharper and more deterministic target distribution, making the future generation direction easier to anticipate and exploit through

---

EAGLE-LLaMA3-Instruct-8B
[3] https://github.com/SafeAILab/EAGLE

*Table 1.* `Llama3.2-3B-Instruct` comparison on SpecBench tasks across temperature=$\{0.0, 0.7, 1.0\}$ and draft nodes=$\{30, 60\}$. WRIT=writing, QA=question-answer, SUMMAR=summarization, TRANS=translation, CODE=coding, M/R=math/reasoning. Both metrics are higher the better.

| Method | Nodes | T | WRIT | | RP | | CODE | | EXT | | STEM | | HUM | | SUMM | | TRAN | | QA | | RAG | | M/R | | AVG | |
|---|---|---|---|---|---|---|---|---|---|---|---|---|---|---|---|---|---|---|---|---|---|---|---|---|---|---|
| | | | $\tau$ | SR | $\tau$ | SR | $\tau$ | SR | $\tau$ | SR | $\tau$ | SR | $\tau$ | SR | $\tau$ | SR | $\tau$ | SR | $\tau$ | SR | $\tau$ | SR | $\tau$ | SR | $\tau$ | SR |
| Eagle3 | 30 | 0.0 | 4.00 | 1.97 | 3.54 | 1.73 | 5.13 | 2.54 | 4.38 | 2.07 | 4.26 | 2.10 | 4.04 | 1.97 | 3.50 | 1.71 | 3.64 | 1.79 | 3.58 | 1.72 | 4.07 | 1.37 | 4.59 | 2.06 | 4.00 | 1.83 |
| ConFu | 30 | 0.0 | **4.55** | **2.18** | **4.18** | **1.96** | **5.72** | **2.75** | **4.96** | **2.27** | **4.85** | **2.33** | **4.59** | **2.18** | **3.95** | **1.88** | **4.16** | **1.98** | **3.91** | **1.83** | **4.52** | **2.00** | **5.21** | **2.45** | **4.41** | **2.11** |
| Eagle3 | 30 | 0.7 | 3.78 | 1.78 | 3.41 | 1.60 | 4.86 | 2.30 | 4.17 | 1.90 | 4.00 | 1.89 | 3.75 | 1.75 | 3.44 | 1.60 | 3.40 | 1.60 | 3.43 | 1.60 | 3.90 | 1.29 | 4.32 | 2.01 | 3.81 | 1.70 |
| ConFu | 30 | 0.7 | **4.24** | **1.97** | **3.99** | **1.78** | **5.43** | **2.54** | **4.77** | **2.09** | **4.61** | **2.15** | **4.15** | **1.93** | **3.82** | **1.8** | **3.95** | **1.8** | **3.75** | **1.7** | **4.42** | **1.86** | **4.86** | **2.24** | **4.29** | **1.95** |
| Eagle3 | 30 | 1.0 | 3.23 | 1.51 | 2.90 | 1.37 | 4.47 | 2.12 | 4.08 | 1.85 | 3.24 | 1.54 | 3.19 | 1.52 | 3.22 | 1.51 | 3.03 | 1.43 | 3.09 | 1.43 | 3.44 | 1.52 | 3.97 | 1.85 | 3.58 | 1.59 |
| ConFu | 30 | 1.0 | **3.62** | **1.72** | **3.38** | **1.59** | **5.11** | **2.44** | **4.17** | **1.89** | **3.49** | **1.67** | **3.45** | **1.63** | **3.55** | **1.66** | **3.54** | **1.68** | **3.39** | **1.58** | **3.83** | **1.73** | **4.41** | **2.06** | **3.82** | **1.78** |
| Eagle3 | 60 | 0.0 | 4.30 | 2.11 | 3.86 | 1.88 | 5.41 | 2.68 | 4.61 | 2.18 | 4.53 | 2.24 | 4.34 | 2.13 | 3.69 | 1.80 | 3.88 | 1.89 | 3.81 | 1.83 | 4.29 | 1.95 | 4.88 | 2.37 | 4.25 | 2.05 |
| ConFu | 60 | 0.0 | **4.90** | **2.29** | **4.42** | **2.09** | **5.96** | **2.88** | **5.13** | **2.37** | **5.16** | **2.49** | **4.83** | **2.31** | **4.18** | **1.99** | **4.40** | **2.10** | **4.17** | **1.96** | **4.75** | **2.10** | **5.48** | **2.59** | **4.76** | **2.24** |
| Eagle3 | 60 | 0.7 | 4.05 | 1.89 | 3.77 | 1.75 | 5.01 | 2.35 | 4.58 | 2.06 | 4.20 | 1.97 | 3.91 | 1.83 | 3.56 | 1.65 | 3.65 | 1.69 | 3.69 | 1.68 | 4.16 | 1.81 | 4.63 | 2.13 | 4.06 | 1.86 |
| ConFu | 60 | 0.7 | **4.64** | **2.11** | **4.25** | **1.90** | **5.45** | **2.48** | **5.01** | **2.18** | **4.78** | **2.18** | **4.49** | **2.03** | **4.03** | **1.80** | **4.18** | **1.89** | **4.05** | **1.80** | **4.57** | **1.96** | **5.20** | **2.32** | **4.54** | **2.02** |
| Eagle3 | 60 | 1.0 | 3.47 | 1.67 | 3.08 | 1.48 | 4.37 | 2.11 | 4.26 | 1.97 | 3.36 | 1.62 | 3.15 | 1.51 | 3.37 | 1.60 | 3.28 | 1.56 | 3.34 | 1.57 | 3.62 | 1.65 | 4.12 | 1.95 | 3.60 | 1.70 |
| ConFu | 60 | 1.0 | **3.86** | **1.73** | **3.56** | **1.59** | **5.09** | **2.29** | **4.49** | **1.94** | **4.18** | **1.88** | **3.42** | **1.52** | **3.81** | **1.69** | **3.74** | **1.67** | **3.63** | **1.59** | **4.08** | **1.73** | **4.76** | **2.10** | **4.08** | **1.80** |

*Table 2.* `Llama3.1-8B-Instruct` comparison on SpecBench tasks across temperature=$\{0.0, 0.7, 1.0\}$ and draft nodes=$\{30, 60\}$. Both metrics are higher the better.

| Method | Nodes | T | WRIT | | RP | | CODE | | EXT | | STEM | | HUM | | SUMM | | TRAN | | QA | | RAG | | M/R | | AVG | |
|---|---|---|---|---|---|---|---|---|---|---|---|---|---|---|---|---|---|---|---|---|---|---|---|---|---|---|
| | | | $\tau$ | SR | $\tau$ | SR | $\tau$ | SR | $\tau$ | SR | $\tau$ | SR | $\tau$ | SR | $\tau$ | SR | $\tau$ | SR | $\tau$ | SR | $\tau$ | SR | $\tau$ | SR | $\tau$ | SR |
| Eagle3 | 30 | 0.0 | 4.39 | 2.43 | 4.26 | 2.31 | 5.65 | 3.15 | 4.89 | 2.59 | 5.02 | 2.80 | 4.60 | 2.59 | 3.76 | 2.04 | 4.28 | 2.35 | 4.35 | 2.32 | 4.78 | 1.59 | 5.16 | 2.82 | 4.59 | 2.36 |
| ConFu | 30 | 0.0 | **5.03** | **2.74** | **4.64** | **2.50** | **6.00** | **3.32** | **5.19** | **2.75** | **5.47** | **3.03** | **5.18** | **2.89** | **4.07** | **2.20** | **4.87** | **2.66** | **4.63** | **2.46** | **5.19** | **2.60** | **5.65** | **3.08** | **5.01** | **2.69** |
| Eagle3 | 30 | 0.7 | 4.09 | 2.23 | 3.91 | 2.14 | 5.36 | 2.95 | 4.72 | 2.48 | 4.45 | 2.45 | 4.27 | 2.34 | 3.64 | 1.96 | 4.03 | 2.19 | 3.99 | 2.12 | 4.49 | 2.27 | 4.88 | 2.63 | 4.31 | 2.31 |
| ConFu | 30 | 0.7 | **4.57** | **2.42** | **4.33** | **2.25** | **5.74** | **3.07** | **5.07** | **2.59** | **4.77** | **2.67** | **4.77** | **2.57** | **3.99** | **2.08** | **4.16** | **2.15** | **4.49** | **2.36** | **4.84** | **2.36** | **5.34** | **2.81** | **4.70** | **2.44** |
| Eagle3 | 30 | 1.0 | 3.35 | 1.81 | 3.34 | 1.80 | 4.84 | 2.63 | 4.42 | 2.29 | 4.01 | 2.19 | 3.02 | 1.65 | 3.30 | 1.75 | 3.64 | 1.95 | 3.57 | 1.88 | 3.96 | 1.99 | 4.48 | 2.41 | 3.85 | 2.04 |
| ConFu | 30 | 1.0 | **4.18** | **2.16** | **3.69** | **1.96** | **5.05** | **2.71** | **4.54** | **2.31** | **3.91** | **2.11** | **3.65** | **1.97** | **3.60** | **1.88** | **4.09** | **2.17** | **3.46** | **1.81** | **4.32** | **2.13** | **4.71** | **2.49** | **4.11** | **2.15** |
| Eagle3 | 60 | 0.0 | 4.74 | 2.59 | 4.50 | 2.43 | 5.91 | 3.27 | 5.04 | 2.65 | 5.29 | 2.94 | 4.92 | 2.75 | 3.96 | 2.14 | 4.58 | 2.50 | 4.69 | 2.48 | 5.03 | 2.52 | 5.49 | 2.99 | 4.87 | 2.61 |
| ConFu | 60 | 0.0 | **5.31** | **2.79** | **4.92** | **2.59** | **6.18** | **3.32** | **5.34** | **2.73** | **5.72** | **3.09** | **5.45** | **2.95** | **4.24** | **2.23** | **5.15** | **2.73** | **4.91** | **2.53** | **5.37** | **2.60** | **5.89** | **3.11** | **5.25** | **2.73** |
| Eagle3 | 60 | 0.7 | 4.46 | 2.31 | 4.23 | 2.19 | 5.51 | 2.90 | 4.91 | 2.45 | 4.81 | 2.52 | 4.67 | 2.46 | 3.78 | 1.94 | 4.33 | 2.24 | 4.33 | 2.21 | 4.79 | 2.28 | 5.26 | 2.70 | 4.61 | 2.34 |
| ConFu | 60 | 0.7 | **5.16** | **2.62** | **4.65** | **2.37** | **5.90** | **3.04** | **5.13** | **2.53** | **5.35** | **2.75** | **5.11** | **2.65** | **4.05** | **2.03** | **4.85** | **2.46** | **4.49** | **2.24** | **5.07** | **2.38** | **5.61** | **2.83** | **4.96** | **2.48** |
| Eagle3 | 60 | 1.0 | 3.87 | 2.03 | 3.49 | 1.82 | 4.68 | 2.48 | 4.41 | 2.20 | 4.00 | 2.12 | 3.60 | 1.92 | 3.57 | 1.83 | 3.97 | 2.07 | 3.75 | 1.91 | 4.13 | 2.01 | 4.73 | 2.46 | 4.07 | 2.09 |
| ConFu | 60 | 1.0 | **4.48** | **2.26** | **4.00** | **2.02** | **5.34** | **2.73** | **4.85** | **2.36** | **4.18** | **2.14** | **4.01** | **2.05** | **3.79** | **1.89** | **4.34** | **2.19** | **3.77** | **1.88** | **4.40** | **2.09** | **5.04** | **2.54** | **4.37** | **2.18** |

contemplate signals.

**Effect of Draft Tree Budget.** We further observe that ConFu provides consistent improvements under both 30-node and 60-node draft trees, representing different budgets for speculative decoding. At the same time, inserting contemplate tokens introduces additional computation proportional to the number of draft nodes. This motivates future work on leveraging the robustness of future prediction to reduce the number of contemplation tokens required during inference, further improving scalability.

Overall, ConFu yields efficiency improvements over EAGLE-3 across model scales and decoding settings while incurring minimal memory overhead as shown in Appendix Table 5. For Llama3.2-3B-Instruct, ConFu increases average acceptance length by approximately by an average of $1.11\times$ and improves SR by roughly $8.2\%$ across temperatures and node configurations for EAGLE-3. Similar trends hold for Llama3.1-8B-Instruct, where ConFu consistently achieves higher accept length and speedups compared to EAGLE-3. These results demonstrate that incorporating future-aware contemplate signals effectively mitigates error accumulation in draft models and pushes speculative decoding closer to its full efficiency potential.

### 4.2. Ablation Studies

In this section, we report the ablation studies of ConFu. We evaluate the benefits of using dynamic contemplate tokens with MoE (Section 3.2) and future prediction replication (Section 3.3). We compare ConFu with two of its variants: ConFu without MoE and ConFu without MoE or replication. The results are shown in Table 4. We additionally compare the sensitivity of ConFu to training draft-length, and comparison of ConFu with Eagle3 on a variant of QWen3 model family

**Effect of Dynamic Contemplate Tokens with MoE** Comparing the performance of ConFu without MoE or replication, we observe that by adding the future prediction replication, the average accept length increases about 0.17. It suggests that the robust training strategy improves the effectiveness of future prediction as designed.

**Effect of Robust Training with Future Prediction Replication** Additionally, comparing the results of ConFu and ConFu with MoE, we observe that, by making the contemplate tokens dynamic with MoE, the accept length increases by 0.05 and the speed-up ratio increases by 0.02. It demonstrates the advantage of our proposed dynamic contemplate

*Table 3.* `Qwen3-4B` comparison on SpecBench tasks across temperature=$\{0.0, 0.7, 1.0\}$ and draft nodes=$\{30, 60\}$. Both metrics are higher the better.

| Method | Nodes | T | WRIT | | RP | | CODE | | EXT | | STEM | | HUM | | SUMM | | TRAN | | QA | | RAG | | M/R | | AVG | |
|---|---|---|---|---|---|---|---|---|---|---|---|---|---|---|---|---|---|---|---|---|---|---|---|---|---|---|
| | | | $\tau$ | S/R | $\tau$ | S/R | $\tau$ | S/R | $\tau$ | S/R | $\tau$ | S/R | $\tau$ | S/R | $\tau$ | S/R | $\tau$ | S/R | $\tau$ | S/R | $\tau$ | S/R | $\tau$ | S/R | $\tau$ | S/R |
| Eagle3 | 30 | 0.0 | 3.69 | 2.53 | 3.14 | 2.18 | 4.03 | 2.88 | 4.05 | 2.88 | 3.83 | 2.83 | 3.45 | 2.52 | 3.05 | 2.05 | 3.16 | 2.27 | 3.37 | 2.41 | 3.56 | 2.18 | 4.21 | 2.80 | 3.57 | 2.44 |
| ConFu | 30 | 0.0 | **4.57** | **3.10** | **3.69** | **2.53** | **4.98** | **3.53** | **4.92** | **3.39** | **4.49** | **3.29** | **4.08** | **2.95** | **3.59** | **2.34** | **3.86** | **2.74** | **4.01** | **2.83** | **4.37** | **3.05** | **5.15** | **3.59** | **4.32** | **3.00** |
| Eagle3 | 30 | 0.7 | 3.61 | 2.46 | 3.22 | 2.22 | 3.96 | 2.78 | 4.24 | 2.89 | 3.65 | 2.68 | 3.41 | 2.46 | 3.05 | 2.03 | 3.12 | 2.22 | 3.35 | 2.36 | 3.56 | 2.50 | 4.19 | 2.91 | 3.56 | 2.48 |
| ConFu | 30 | 0.7 | **4.43** | **3.03** | **3.60** | **2.50** | **4.93** | **3.50** | **4.98** | **3.48** | **4.39** | **3.24** | **4.04** | **2.94** | **3.63** | **2.38** | **3.76** | **2.69** | **3.88** | **2.76** | **4.37** | **3.06** | **5.15** | **3.60** | **4.28** | **2.99** |
| Eagle3 | 30 | 1.0 | 3.60 | 2.18 | 3.13 | 2.16 | 3.93 | 2.79 | 4.23 | 2.95 | 3.70 | 2.70 | 3.27 | 2.37 | 3.09 | 2.06 | 3.10 | 2.21 | 3.26 | 2.31 | 3.52 | 2.47 | 4.18 | 2.93 | 3.53 | 2.46 |
| ConFu | 30 | 1.0 | **4.29** | **2.86** | **3.43** | **2.34** | **4.81** | **3.38** | **4.99** | **3.42** | **4.26** | **3.08** | **3.69** | **2.63** | **3.59** | **2.32** | **3.77** | **2.65** | **3.79** | **2.64** | **4.28** | **2.94** | **5.13** | **3.52** | **4.21** | **2.89** |
| Eagle3 | 60 | 0.0 | 3.86 | 2.68 | 3.37 | 2.36 | 4.28 | 3.10 | 4.24 | 3.03 | 3.71 | 2.72 | 3.22 | 2.16 | 3.35 | 2.43 | 3.53 | 2.40 | 3.64 | 2.63 | 3.83 | 2.54 | 4.50 | 3.22 | 3.81 | 2.69 |
| ConFu | 60 | 0.0 | **4.85** | **3.31** | **3.91** | **2.71** | **5.26** | **3.73** | **5.05** | **3.48** | **4.89** | **3.60** | **4.37** | **3.16** | **3.80** | **2.45** | **4.12** | **2.93** | **4.31** | **3.04** | **4.69** | **3.27** | **5.46** | **3.80** | **4.60** | **3.20** |
| Eagle3 | 60 | 0.7 | 3.67 | 1.91 | 3.40 | 1.77 | 4.29 | 2.96 | 4.40 | 2.99 | 3.99 | 2.85 | 3.64 | 2.57 | 3.21 | 2.08 | 3.28 | 2.28 | 3.62 | 2.50 | 3.85 | 2.63 | 4.52 | 2.94 | 3.80 | 2.52 |
| ConFu | 60 | 0.7 | **4.68** | **3.07** | **3.97** | **2.64** | **5.26** | **3.59** | **5.20** | **3.46** | **4.78** | **3.36** | **4.37** | **3.05** | **3.77** | **2.36** | **4.05** | **2.78** | **4.20** | **2.86** | **4.60** | **3.08** | **5.44** | **3.65** | **4.55** | **3.05** |
| Eagle3 | 60 | 1.0 | 3.69 | 2.48 | 3.39 | 2.30 | 4.20 | 2.94 | 4.53 | 3.12 | 3.91 | 2.82 | 3.48 | 2.47 | 3.21 | 2.09 | 3.27 | 2.30 | 3.48 | 2.45 | 3.81 | 2.66 | 4.49 | 3.11 | 3.76 | 2.60 |
| ConFu | 60 | 1.0 | **4.49** | **2.99** | **3.91** | **2.63** | **5.07** | **3.54** | **5.12** | **3.50** | **4.63** | **3.33** | **4.15** | **2.93** | **3.77** | **2.40** | **4.00** | **2.78** | **4.08** | **2.80** | **4.55** | **3.12** | **5.40** | **3.69** | **4.48** | **3.05** |

token mechanism.

**Effect of ConFu on Qwen3 model family** The result in Table 3 demonstrate that ConFu consistently outperforms Eagle3 across all configurations of draft nodes and temperature settings. On average, ConFu achieves a 21.0% improvement in average accepted tokens ($\tau$) and a 19.8% improvement in speed-up ratio (S/R) compared to Eagle3. The performance gap is particularly pronounced with 60 draft nodes at temperature 0.0, where ConFu reaches a $\tau$ of 4.60 versus Eagle3's 3.81, representing a 20.7% improvement. Similarly, the speed-up ratio increases from 2.69 to 3.20, showing an 18.9% gain in efficiency. Even at higher temperatures, which typically introduce more variability, ConFu maintains its advantage with improvements ranging from 19.0% - 21.3% and corresponding speed-up ratio enhancements between 17.3% and 21.0%.

**Effect of training draft length** We ablated over two lengths apart from the default training length=7. We chose a shorter length=4 and a longer length=10 and trained the Llama3.2-3B-Instruct model. Results are provided in Appendix Table 6 where we observe that on average different lengths perform similar thus showing that ConFu is robust to training length.

**Stress Test on Long Drafts: Tail Acceptance Analysis.** While average accepted length ($\tau$) summarizes typical acceleration behavior, it does not capture how often long draft sequences are fully or near-fully accepted—a regime that can disproportionately impact practical speed-ups. To probe this tail behavior, we conduct a stress test with a longer draft length (DL=12) under deterministic decoding (Topk = 1).

Instead of reporting only mean statistics, we analyze the distribution of accepted lengths and visualize the corresponding survival function, i.e., the probability that at least $l$ draft tokens are accepted. As shown in Appendix Figure 5, ConFu exhibits substantially higher tail acceptance than EAGLE-3 across the entire range of $l$, indicating that ConFu more consistently accepts longer draft trajectories under strict

decoding. This result complements the improvements in average $\tau$ and speed-up ratio (S/R) reported in the main experiments, and provides additional evidence that ConFu's gains stem not only from higher average acceptance, but from improved robustness in the long-draft regime.

## 5. Related Work

There is a large body of work on accelerating large language model (LLM) inference. Representative directions include model-wise optimizations such as quantization (Lin et al., 2024; Liu et al., 2025), pruning (Ma et al., 2023), and distillation (Hinton et al., 2015), as well as input-wise techniques such as KV cache compression and pruning (Park et al., 2025; Goel et al., 2025; Xiao et al., 2024). Other approaches explore alternative architectures beyond standard Transformers. While these methods can substantially reduce inference latency or memory usage, they typically incur a degradation in downstream task performance or require additional retraining and careful hyperparameter tuning. In contrast, speculative decoding offers a unique advantage: it can accelerate inference while provably preserving the original sampling distribution of the target model, thereby avoiding any compromise in downstream performance.

Early speculative decoding methods (Leviathan et al., 2023) adopt a linear verification scheme, where a draft model proposes a sequence of tokens and the target model verifies them in parallel, accepting the draft prefix until the first rejection. Subsequent work has focused on improving the efficiency of speculative decoding by refining the drafting and verification procedures. In particular, tree-structured speculative decoding methods (Miao et al., 2024; Sun et al., 2023; Chen et al., 2024) expand the draft space into a tree and verify multiple candidate continuations simultaneously, thereby increasing the expected accepted length per iteration. More recently, (Qin et al., 2025b;a) demonstrate that speculative decoding can be leveraged not only to improve efficiency, but also to enhance output quality. Importantly, these methods primarily operate at the algorithmic level,

*Table 4.* `Llama3.1-8B-Instruct` ablation comparison on SpecBench tasks across temperature=$\{0.0, 0.7\}$ and draft nodes=$\{30\}$. WRIT=writing, QA=question-answer, SUMMAR=summarization, TRANS=translation, CODE=coding, M/R=math/reasoning. Both metrics are higher the better. **Bold numbers** indicate best performance under that temperature.

| Method | T | WRIT | | RP | | CODE | | EXT | | STEM | | HUM | | SUMM | | TRAN | | QA | | RAG | | M/R | | AVG | |
|---|---|---|---|---|---|---|---|---|---|---|---|---|---|---|---|---|---|---|---|---|---|---|---|---|---|
| | | $\tau$ | SR | $\tau$ | SR | $\tau$ | SR | $\tau$ | SR | $\tau$ | SR | $\tau$ | SR | $\tau$ | SR | $\tau$ | SR | $\tau$ | SR | $\tau$ | SR | $\tau$ | SR | $\tau$ | SR |
| Eagle3 | 0.0 | 4.39 | 2.43 | 4.26 | 2.31 | 5.65 | 3.15 | 4.89 | 2.59 | 5.02 | 2.80 | 4.60 | 2.59 | 3.76 | 2.04 | 4.28 | 2.35 | 4.35 | 2.32 | 4.78 | 1.59 | 5.16 | 2.82 | 4.59 | 2.36 |
| ConFu | 0.0 | **5.03** | **2.74** | **4.64** | **2.50** | **6.00** | **3.32** | 5.19 | 2.75 | **5.47** | 3.03 | **5.18** | **2.89** | **4.07** | **2.20** | **4.87** | **2.66** | 4.63 | **2.46** | **5.19** | **2.60** | **5.65** | **3.08** | **5.01** | **2.69** |
| −MoE | 0.0 | 4.97 | 2.73 | 4.50 | 2.44 | 5.94 | 3.30 | **5.23** | **2.76** | 5.46 | **3.04** | 5.06 | 2.84 | 4.04 | 2.19 | 4.83 | 2.65 | 4.60 | **2.46** | 5.12 | 2.57 | 5.59 | 3.05 | 4.97 | 2.67 |
| − MoE & Replicate | 0.0 | 4.77 | 2.59 | 4.37 | 2.36 | 5.82 | 3.22 | 5.07 | 2.69 | 5.19 | 2.88 | 4.81 | 2.69 | 3.96 | 2.13 | 4.59 | 2.5 | 5.49 | 2.38 | 4.90 | 2.22 | 5.49 | 2.98 | 4.81 | 2.55 |
| Eagle3 | 0.7 | 4.09 | 2.23 | 3.91 | 2.14 | 5.36 | 2.95 | 4.72 | 2.48 | 4.45 | 2.45 | 4.27 | 2.34 | 3.64 | 1.96 | 4.03 | 2.19 | 3.99 | 2.12 | 4.49 | 2.27 | 4.88 | 2.63 | 4.31 | 2.31 |
| ConFu | 0.7 | **4.57** | **2.42** | **4.33** | **2.25** | **5.74** | **3.07** | **5.07** | **2.59** | 4.99 | **2.67** | 4.70 | **2.57** | **3.99** | **2.08** | **4.53** | **2.39** | 4.16 | **2.15** | **4.84** | **2.36** | **5.34** | **2.81** | **4.70** | **2.44** |
| −MoE | 0.7 | 4.55 | 2.37 | 4.19 | 2.19 | 5.53 | 2.91 | 4.93 | 2.47 | **5.01** | 2.64 | **4.73** | 2.51 | 3.88 | 1.99 | 4.45 | 2.31 | **4.18** | 2.12 | 4.83 | 2.32 | 5.27 | 2.72 | 4.64 | 2.37 |
| − MoE & Replicate | 0.7 | 4.37 | 2.32 | 4.05 | 2.16 | 5.50 | 2.94 | 4.93 | 2.52 | 4.64 | 2.50 | 4.49 | 2.43 | 3.82 | 2.00 | 4.25 | 2.25 | 4.05 | 2.11 | 4.64 | 2.28 | 5.23 | 2.76 | 4.51 | 2.36 |

modifying the drafting and verification strategy while remaining agnostic to the specific architectures of the draft and target models. As a result, they are orthogonal to approaches that focus on improving the draft model itself.

Since the effectiveness of speculative decoding strongly depends on the quality of the draft model, a parallel line of work investigates more powerful drafting architectures. Early speculative decoding frameworks rely on standalone small models as drafters. Medusa (Cai et al., 2024) improves upon this paradigm by attaching multiple lightweight prediction heads to the target model, enabling the parallel generation of future tokens. Notably, the **EAGLE** family (Li et al., 2024a;b; 2025) represents the current state of the art in draft-model design for speculative decoding. EAGLE-1 (Li et al., 2024a) introduces a single-layer Transformer that reuses the target model's key–value cache to autoregressively predict future tokens at the feature level. EAGLE-2 (Li et al., 2024b) further incorporates context-aware dynamic draft trees to adaptively balance exploration and verification. HASS (Zhang et al., 2025) and Griffin (Hu et al., 2025) aims to address the mismatch between training and inference of EAGLE by modifying its training strategy. EAGLE-3 (Li et al., 2025) significantly advances both model architecture and training methodology, achieving new records in speculative decoding throughput. Across a wide range of benchmarks, the EAGLE models consistently outperform prior draft models (Cai et al., 2024; Zhang et al., 2025), and are widely regarded as the strongest draft-model-based speculative decoding approach to date.

## 6. Conclusion

In this work, we introduced ConFu, a new speculative decoding framework that improves draft model quality by capturing target model's current "thought". By leveraging contemplate tokens and soft prompts, ConFu allows the draft model to access lightweight, future-oriented signals from the target model at negligible inference cost. We further proposed a dynamic contemplate token mechanism based on a Mixture-of-Experts architecture, which adapts the future prediction to diverse generation contexts, and a robust training framework that learns stable future representations

through anchor token sampling and prediction replication. Extensive experiments on SpecBench with strong target models demonstrate that ConFu consistently improves token acceptance rates and inference efficiency over the state-of-the-art EAGLE-3 baseline across a wide range of tasks and decoding configurations. These results suggest that equipping draft models with future-aware signals is an effective way to mitigate error accumulation and improves speculative decoding's effectiveness. More broadly, ConFu highlights the importance of modeling high-level generation intent in speculative decoding. We believe this perspective opens new avenues for improving inference efficiency by bridging latent reasoning with speculative decoding.

## Impact Statement

This work contributes to the growing body of research on efficient large language model inference. By improving the effectiveness of speculative decoding without modifying or fine-tuning the target model, ConFu enables faster text generation with reduced computational cost and energy consumption. This has positive implications for deploying large language models in resource-constrained environments, such as real-time systems, edge devices, and large-scale serving infrastructures, where inference efficiency is a critical concern.

ConFu does not introduce new model capabilities beyond those of the underlying target language model, nor does it alter the sampling distribution of the target model. As a result, it does not raise new risks related to model misuse, bias amplification, or content safety beyond those already present in existing language models. The framework is designed as an inference-time optimization and is orthogonal to issues of data collection, model alignment, and training-time bias.

Overall, we view ConFu as a systems-level contribution that helps make large language models more accessible and sustainable, while preserving their original behavior and safety characteristics.

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

## A. Confu Overhead Compared to Eagle3

| Model (60 nodes) | Eagle3 (GB) | ConFu (GB) |
|---|---|---|
| Llama-3.2-3B-Instruct | 7.50 | 7.52 |
| Llama-3.1-8B-Instruct | 17.08 | 17.12 |

*Table 5.* **Peak GPU memory usage** (GB) when running with 60 draft nodes. Results are averaged over 10 SpecBench samples.

## B. Ablation Results

*Table 6.* Llama3.2-3B-Instruct ConFu average acceptance rate ($\tau$, higher is better) across temperature=$\{0.0, 0.7, 1.0\}$ and draft nodes=$\{30, 60\}$, for training block lengths $\{4,7,10\}$. Bold indicates the best length for each (Nodes, T) setting. WRIT=writing, QA=question-answer, SUMMAR=summarization, TRANS=translation, CODE=coding, M/R=math/reasoning

| Length | Nodes | T | WRIT | RP | CODE | EXT | STEM | HUM | SUMM | TRAN | QA | RAG | M/R | AVG |
|---|---|---|---|---|---|---|---|---|---|---|---|---|---|---|
| 4 | 30 | 0.0 | **4.72** | 4.10 | 5.70 | 4.95 | **4.85** | 4.48 | **4.17** | 3.93 | 3.88 | 4.56 | 5.16 | 4.50 |
| 7 | 30 | 0.0 | 4.55 | **4.18** | 5.72 | 4.96 | **4.85** | **4.59** | 3.95 | **4.16** | **3.91** | 4.52 | **5.21** | 4.41 |
| 10 | 30 | 0.0 | 4.64 | 4.02 | **5.76** | **4.99** | 4.79 | 4.51 | **4.17** | 3.96 | **3.91** | **4.57** | 5.15 | **4.50** |
| 4 | 30 | 0.7 | **4.35** | 3.87 | 5.37 | 4.67 | 4.55 | **4.30** | **3.88** | 3.78 | 3.70 | 4.35 | **4.94** | 4.27 |
| 7 | 30 | 0.7 | 4.24 | **3.99** | 5.43 | **4.77** | **4.61** | 4.15 | 3.82 | **3.95** | **3.75** | **4.42** | 4.86 | **4.29** |
| 10 | 30 | 0.7 | **4.35** | 3.97 | **5.62** | 4.74 | 4.59 | 4.22 | **3.88** | 3.81 | 3.63 | 4.34 | 4.86 | 4.26 |
| 4 | 30 | 1.0 | 3.59 | **3.51** | 4.72 | 4.35 | **3.78** | 3.52 | 3.45 | 3.53 | 3.40 | **3.93** | **4.50** | **3.85** |
| 7 | 30 | 1.0 | **3.62** | 3.38 | **5.11** | 4.17 | 3.49 | 3.45 | **3.55** | **3.54** | 3.39 | 3.83 | 4.41 | 3.82 |
| 10 | 30 | 1.0 | 3.52 | 3.33 | 5.08 | **4.37** | 3.77 | 3.26 | 3.49 | 3.49 | **3.41** | 3.90 | 4.33 | 3.80 |
| 4 | 60 | 0.0 | 4.96 | 4.38 | 5.90 | 5.15 | **5.17** | 4.71 | **4.38** | 4.13 | **4.15** | 4.75 | 5.42 | 4.73 |
| 7 | 60 | 0.0 | 4.90 | **4.42** | **5.96** | 5.13 | 5.16 | **4.83** | 4.18 | **4.40** | **4.17** | 4.75 | **5.48** | 4.76 |
| 10 | 60 | 0.0 | **4.97** | 4.32 | 5.88 | **5.19** | 5.10 | 4.80 | **4.38** | 4.16 | 4.13 | **4.75** | 5.42 | 4.73 |
| 4 | 60 | 0.7 | 4.59 | 3.94 | **5.61** | 4.90 | **4.85** | **4.61** | 4.13 | 4.04 | 4.03 | **4.65** | 5.19 | **4.54** |
| 7 | 60 | 0.7 | **4.64** | **4.25** | 5.45 | **5.01** | 4.78 | 4.49 | 4.03 | **4.18** | **4.05** | 4.57 | **5.20** | **4.54** |
| 10 | 60 | 0.7 | 4.59 | 4.13 | 5.58 | 4.87 | 4.79 | 4.49 | **4.15** | 3.95 | 4.04 | 4.56 | 5.18 | 4.50 |
| 4 | 60 | 1.0 | 3.93 | **3.70** | 4.71 | **4.69** | 4.04 | 3.73 | 3.58 | **3.79** | 3.55 | **4.20** | **4.82** | **4.08** |
| 7 | 60 | 1.0 | 3.86 | 3.56 | **5.09** | 4.49 | **4.18** | 3.42 | **3.81** | 3.74 | **3.63** | 4.08 | 4.76 | **4.08** |
| 10 | 60 | 1.0 | **4.02** | 3.58 | 4.93 | 4.57 | 3.80 | **3.87** | 3.71 | 3.71 | 3.59 | **4.20** | 4.75 | **4.08** |

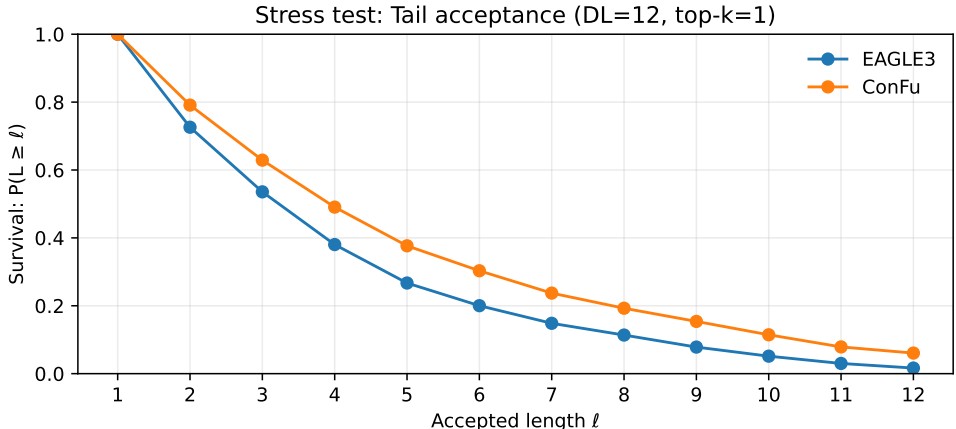

*Figure 5.* Survival function of accepted draft length, showing the probability that at least $l$ consecutive draft tokens are accepted. ConFu consistently exhibits higher tail acceptance than EAGLE-3, indicating more robust acceptance of long draft trajectories under strict decoding.

