# OpenReview forum: "ConFu: Contemplate the Future for Better Speculative Sampling"
_ICML.cc/2026/Conference — ICML 2026 regular_

### Official Review · Reviewer_NAyw · 2026-03-09

**Soundness:** 2
**Presentation:** 3
**Significance:** 2
**Originality:** 3
**Overall Recommendation:** 4
**Confidence:** 4

**Summary:**

The paper introduces ConFu, which proposes multiple improvements to EAGLE, a highly performant speculative decoding framework, with the effect of improving acceptance rates and generation speeds by 8-11% on SpecBench across diverse generation domains.

Specifically, the paper introduces "contemplate tokens", which are appended to the main target model at each step, and which can attend to special soft prompts prepended to the input of the target model. Instead of a fixed learnt soft prompt, the input embeddings for the "contemplate tokens" are mapped via an MoE from the last accepted hidden state. Finally, the authors show how to train this architecture in an efficient manner by "replicating" the speculation process at specific positions and making the "contemplate tokens" more robust to positional offsets.

**Compliance With Llm Reviewing Policy:**

Affirmed.

**Final Justification:**

The rebuttal addressed my concerns and while it would benefit from further empirical validation, overall I find the work quite neat.

**Key Questions For Authors:**

n/a

**Limitations:**

yes

**Strengths And Weaknesses:**

### Strengths

[S1] The measured speed ups is strong across a range of benchmarks on SpecBench across two model sizes.

[S2] The authors are careful to make sure that the original target model prompt is not affected by the soft prompt tokens or the new contemplate token, thus ensuring that speculation remains faithful to the original target model. The idea of the "dynamic contemplate token" also re-uses last hidden states that would already be available at inference time with speculative decoding, introducing a kind of recurrence. I also find the soft prompts that are only attended to by the contemplate tokens quite a clever idea to inject extra capacity to the contemplation that is shared across the sequence (e.g. as opposed to always injecting multiple contemplate tokens)

[S3] The authors make sure that the training setting carefully matches the inference time setting and make reasonable compromises to ensure training efficiency, i.e. only "simulating" the speculation at a few positions.

### Weaknesses

[W1] The concatenation of the contemplate token at every generation step *should* introduce a substantial overhead, and I am suspicious of the papers claim that this is negligible, given that the the average speculation length is often just 4-5 tokens on SpecBench. Therefore, I think it is critical to see a more detailed breakdown of how this method balances the improved performance and speed drawbacks. I think it's also important to "model" how this balance would look like on bigger models, where the "FLOP cost" of appending an additional token is more significant.

[W2] The method is rather complex (between <contemplate tokens>, soft prompt, <f> tokens, MoE input tokens, replicate training) and a greater emphasis could be placed upon how these elements contribute to performance and the latency/cost penalty that they occur. I view most of these elements as injecting extra capacity to the draft head and I think even an analysis of "training validation loss" could be interesting here, too.

[W3] I find Figure 1 misleading, as the paper has little empirical evidence to show that the "contemplate token" mechanism actually leads to better alignment (rather than just providing additional capacity to the draft model).

---

> ### Author Rebuttal · Authors · 2026-03-31
>
> #  I think it's also important to "model" how this balance would look like on bigger models, where the "FLOP cost" of appending an additional token is more significant.
>
> Thank you for the valuable suggestion. We agree that evaluation on larger models (e.g., 70B) is important. Unfortunately, due to resource constraints, we were not able to complete training with a 70B target model which would require more time. Instead, we provide an analysis of the additional cost introduced by ConFu.
>
> **Decoding Stage**
>
> Let $N$ denote the KV cache length, $T$ the number of draft tokens, and $L$ the number of layers. For standard speculative decoding, the verification cost per step can be written as
> $
> [\alpha (N+T)T + \beta T + \gamma]L
> $
> in FLOPs and
> $
> [a(N+T) + b]L
> $
> in memory, where $\alpha, \beta, \gamma, a, b$ are constants.
>
> In ConFu, due to the introduction of contemplate tokens, the number of tokens to verify becomes $2T$. At first glance, this appears to double the verification cost. However, this increase is much less significant in realistic regimes. In practical LLM inference, the KV cache length $N$ is typically on the order of thousands, while $T$ is relatively small (e.g., $\sim$50–100). Therefore, we have
> $$
> N \gg T \quad \Rightarrow \quad N + T \approx N.
> $$
>
> Under this regime, the **dominant memory term remains unchanged**, and the additional contemplate tokens introduce only a negligible increase in memory footprint.
>
> For FLOPs, while the arithmetic cost increases, speculative decoding is well known to be **memory-bandwidth bound rather than compute-bound**. A commonly used approximation (validated in prior works) is that verifying $T$ tokens has comparable latency to generating a single token, i.e.,
> $$
> [\alpha (N+T)T + \beta T + \gamma]L \approx [\alpha (N+1) + \beta + \gamma]L.
> $$
> Since ConFu does not change the asymptotic dependence on $N$ and does not increase the number of target-model forward passes, the additional computation from contemplate tokens has **limited impact on end-to-end latency**.
>
> **Prefilling Stage**
>
> During prefilling, ConFu only introduces a constant number of soft prompt tokens and a single contemplate token. Therefore, the additional cost is $O(1)$ with respect to sequence length and model size, and is negligible in practice.
>
> **Implications for scaling to large models**
>
> The analysis above shows that when input length N is significantly larger than draft node count T, and when memory cost dominates the inference cost of the decoding stage, the contemplate tokens of ConFu introduces minimal extra overhead.
> Importantly, all costs above scale **linearly** with the number of layers $L$. Scaling to larger models (e.g., 70B) primarily increases $L$ (and/or $d$), but does not change the **relative overhead** introduced by ConFu. Therefore, our analysis indicates that ConFu maintains similar efficiency characteristics at larger scales.
>
> **Empirical Evidence**
>
> We also provide peak memory usage when using Eagle-3 vs ConFu to solidify that our method doesn’t increase overhead.The table below we show the peak memory usage.
> | 60 nodes    	| EAGLE3 | Confu |
> |-|-|-|
> | 3b	| 7.50   | 7.52  |
> | 8b 	| 17.08  | 17.12 |
> As you can see, ConFu leads to minimal overheads: 0.02 GB for 3B model and 0.04 for 8B model. For larger models, like 70B, the overhead should also be small.
>
> #  a greater emphasis could be placed upon how these elements contribute to performance and the latency/cost penalty that they occur
>
> Thank you for this valuable suggestion. In Table 3 of our paper, we provide an ablation study showing the effects of MoE input tokens and replicate training. We can see that removing either component leads to a clear degradation in inference efficiency.
> Additionally, the soft prompt tokens are also critical to the effectiveness of ConFu. Without soft prompt tokens, the contemplate token cannot generate meaningful future predictions, and ConFu achieves a similar acceptance rate to EAGLE-3. We will include this ablation (removing soft prompt tokens) in Table 3 to further clarify the contribution of each component.
>
> # Figure 1 is misleading
>
> Thank you for pointing this out. To address your concern, we provide the following result showing ConFu leads to better alignment in the long horizon. Specifically, we compare the acceptance probability of a single draft sequence with length 12 and measure the empirical acceptance distribution:
>
> Eagle3 acceptance length distribution : (avg BE = 3.44)
>
> [0.274, 0.190, 0.155, 0.113, 0.067, 0.052, 0.035, 0.035, 0.027, 0.021, 0.014, 0.016]
>
> Confu acceptance length distribution: (avg BE = 4.31)
>
> [0.209, 0.162, 0.138, 0.114, 0.074, 0.066, 0.045, 0.039, 0.039, 0.036, 0.018, 0.061]
>
> We can see that ConFu indeed has better alignment for long horizon cases. In the main paper, we will add a bar plot for this ablation

---

> > ### Author Rebuttal · Reviewer_NAyw · 2026-04-04
> >
> > Thank you for the rebuttal and engaging with my concerns. I think including more experimental measurements across intermediate scale is still important but I'm happy to raise my score.

---

> > > ### Author Response · Authors · 2026-04-04
> > >
> > > We thank the reviewer for their thoughtful follow‑up and are glad that our rebuttal addressed their questions. We appreciate the reviewer’s updated evaluation and will reflect their feedback in the final version.

---

### Official Review · Reviewer_gv4u · 2026-03-11

**Soundness:** 3
**Presentation:** 3
**Significance:** 2
**Originality:** 2
**Overall Recommendation:** 3
**Confidence:** 4

**Summary:**

The paper introduces ConFu, which appends a "contemplate token" to the input sequence and prepend learnable soft prompts to the kv cache. The target model's hidden state corresponding to this contemplate token is extracted as the future prediction vector, which is then fed into a lightweight, EAGLE-style draft model as an additional conditioning token. The paper also proposes parameterizing the contemplate and future tokens using a dynamic MoE routing mechanism. To manage memory constraints during training, the authors employ anchor token sampling and reuse the future prediction vector for neighboring tokens. Empirical results on SpecBench show an 8-11% improvement in acceptance rate and speedup ratio over EAGLE-3 on Llama-3 3B and 8B models.

**Compliance With Llm Reviewing Policy:**

Affirmed.

**Final Justification:**

The rebuttal has addressed some of my concerns. Nevertheless, I will keep my scores unchanged due to insufficient novelty and limited overall appeal of this work.

**Key Questions For Authors:**

1. How does ConFu perform compared to EAGLE-3 when the batch size is greater than 1?
2. During inference, how does ConFu's GPU memory overhead compare to that of standard decoding and EAGLE-3?

**Limitations:**

yes

**Strengths And Weaknesses:**

Strengths:
1. Correctly identifies autoregressive drift as a fundamental limitation in prefix-only draft models.
2. Introduces a clever engineering implementation that inserts contemplate tokens during the target model's parallel verification pass, avoiding sequential latency penalties.
3. Provides a credible baseline by comparing directly against EAGLE-3, demonstrating a clear 8-11% improvement.

Weaknesses:
1. The effectiveness of ConFu appears to stem solely from extracting additional information from the target model to provide to the draft model. Although using mechanisms like "pause tokens" to achieve this is somewhat elegant, I think it offers no new insights and further adds significant complexity to the entire speculative sampling system.
2. The experiments in this paper were conducted only with a batch size of 1. In contrast, EAGLE-3, which serves as a baseline in this work, was evaluated across batch sizes ranging from 1 to 64, demonstrating improvements in all cases. Therefore, it remains unclear whether ConFu can maintain its performance advantage in real-world production environments with larger batch sizes.

---

> ### Author Rebuttal · Authors · 2026-03-31
>
> # I think it offers no new insights and further adds significant complexity to the entire speculative sampling system.
>
> Thank you for raising this important concern. We agree that, at a high level, ConFu leverages additional information from the target model to assist the draft model. However, we respectfully disagree that this idea is trivial or lacks new insights. In fact, the core contribution of our work lies precisely in how to extract and utilize this information in a way that is both efficient within speculative decoding constraints and effective in improving draft quality.
>
> ## Conceptual Novelty
> To the best of our knowledge, this is the first work that explicitly introduces the concept of LLM implicit thinking into speculative decoding. Prior approaches (e.g., EAGLE-style methods) are limited to reusing existing hidden states, which primarily encode past context. In contrast, our key insight is that the target model can be prompted to expose its future generation trajectory, providing fundamentally different and more informative signals for drafting. This represents a new direction: moving from passive reuse of internal states to actively eliciting forward-looking signals from the target model.
>
> ## Methodology Novelty
> While the high-level idea may appear simple, making it practical in speculative decoding is highly non-trivial. A naïve implementation, such as inserting pause tokens or performing additional forward passes. either fails to produce meaningful signals or incurs prohibitive overhead.
> Our contributions address these challenges in a principled way:
> - We introduce contemplate tokens and soft prompt tokens that enable the target model to generate future-oriented signals in parallel with verification, avoiding extra inference passes.
> - We show that naïvely adding such tokens is insufficient, and propose dynamic contemplate tokens with MoE (Sec. 3.2) to adaptively produce informative signals.
> - We further design training strategies (Sec. 3.3), including anchor token sampling and future prediction replication, to ensure these signals are both stable and useful for the draft model.
>
> Together, these components form a coherent framework that transforms an intuitive idea into a practical mechanism, which we believe constitutes clear methodological novelty.
>
> ## Empirical Evidence
> If the idea were as straightforward as suggested, one would expect limited or inconsistent improvements. However, our experiments show consistent gains of 8–11% in acceptance length and speed over EAGLE-3, a strong baseline. These improvements demonstrate that our method provides a meaningful advancement in the efficiency–quality tradeoff.
>
> For these reasons, we believe the contribution goes beyond incremental refinement and provides meaningful novelty and insight to the community.
>
> # Experiments with larger batch size
>
> Thank you for this insightful comment. We agree that evaluating performance under larger batch sizes is important for production scenarios, and we appreciate the opportunity to clarify our experimental setting and its implications.
>
> ## Batch Size = 1 Is the Standard Protocol for Draft-Quality Improvements.
> Our evaluation follows the conventional protocol adopted by prior papers that focus on improving the draft model effectiveness, including HASS [1] and EAGLE-2 [2]. These works consistently evaluate under batch size = 1 because the primary goal is to isolate improvements in acceptance rate / draft accuracy, rather than system-level throughput optimizations.
>
> In this context, our comparison with EAGLE-3 is fair and controlled: we evaluate under the same setting to ensure that improvements are attributable to the proposed method, rather than confounded by batching strategies.
>
> ## Orthogonality to Batching and System-Level Optimizations.
> The core contribution of our work is to improve the quality of draft proposals by leveraging future-oriented signals from the target model. This mechanism operates at the token generation level and is independent of how requests are batched at the system level.
>
> ## Practical Application of Batch Size = 1.
> We also note that batch size = 1 is not merely a simplified setting, but is highly relevant in many real-world deployments. For example, in edge inference scenarios (e.g., mobile devices, on-device assistants), requests are processed individually due to latency and memory constraints. There exist Industry efforts that specifically target such single-request settings, where improving per-sequence efficiency is critical.
>
> [1] Zhang, Lefan, et al. "Learning harmonized representations for speculative sampling." arXiv preprint arXiv:2408.15766 (2024).
> [2] Li, Yuhui, et al. "Eagle-2: Faster inference of language models with dynamic draft trees." Proceedings of the 2024 conference on empirical methods in natural language processing. 2024.

---

> > ### Author Rebuttal · Reviewer_gv4u · 2026-04-01
> >
> > Thanks for your rebuttal, which has addressed some of my concerns. Nevertheless, I will keep my scores and weak rejection unchanged due to insufficient novelty and limited overall appeal of this work.

---

> > > ### Author Response · Authors · 2026-04-01
> > >
> > > We thank the reviewer again for their time and feedback
> > >
> > > To add another data point for the reviewer regarding the conceptual novelty
> > >
> > > We stress tested Eagle3 and Confu for predicting single long draft trajectories correctly instead of tree.
> > > We set the draft length = 12.
> > > We observed that with Confu the long tail acceptance has larger mass compared to Eagle3 (0.016 vs 0.61 for draft token at position 12)
> > > We believe this cannot be trivially without significantly increasing the model capacity. In our method we are simply adding future tokens.
> > >
> > > Eagle3 acceptance length distribution : (avg BE = 3.44)
> > >
> > > [0.274, 0.190, 0.155, 0.113, 0.067, 0.052, 0.035, 0.035, 0.027, 0.021, 0.014, 0.016]
> > >
> > > Confu acceptance length distribution: (avg BE = 4.31)
> > >
> > > [0.209, 0.162, 0.138, 0.114, 0.074, 0.066, 0.045, 0.039, 0.039, 0.036, 0.018, 0.061]
> > >
> > > We can see that ConFu indeed has better alignment for long horizon cases.

---

### Official Review · Reviewer_apEV · 2026-03-12

**Soundness:** 2
**Presentation:** 3
**Significance:** 3
**Originality:** 3
**Overall Recommendation:** 4
**Confidence:** 3

**Summary:**

This paper proposes a novel speculative decoding framework named ConFu (Contemplate the Future), which aims to address the error accumulation issue caused by existing draft models relying solely on the current prefix. The method extracts the target model's future generation intent at a minimal inference cost by introducing "contemplate tokens" and "soft prompts" into the target model.
1. It introduces the pause token (contemplate token) mechanism into speculative sampling, utilizing continuous latent representations to guide the draft model's generation.
2. It proposes a dynamic contemplate token mechanism based on a Mixture-of-Experts (MoE) architecture, allowing the prediction of future intent to adaptively fit different contexts.
3. It designs an Anchor Token Sampling and Future Prediction Replication strategy, effectively controlling the memory overhead of long-sequence training and enhancing feature robustness

**Compliance With Llm Reviewing Policy:**

Affirmed.

**Key Questions For Authors:**

Please provide clear responses to the following questions during the Rebuttal phase:
1. Can you provide a detailed hardware metrics table explicitly comparing the Peak VRAM usage, KV Cache size, and absolute latency (wall-clock time) differences between ConFu and EAGLE-3 under the same $T$ node settings (e.g., $T=60$)?
2. For tasks with strict syntactic constraints like Code or Math, does forcibly reusing $f_t$ within a window of length $l$ cause distribution drift? Please provide ablation data for different values of $l$.
3. Currently, the method is only validated up to the 8B level. At the 70B+ model scale, does ConFu's marginal advantage over EAGLE-3 expand or shrink?

**Limitations:**

This paper aims high and elegantly integrates the frontier of latent reasoning (pause tokens) with inference acceleration (speculative decoding), breaking through the pure autoregressive fitting bottleneck of draft models. Its proposed MoE dynamic representation and Anchor sampling training form a logically consistent engineering loop. However, the paper falls short in its transparency regarding VRAM overhead/latency and fails to demonstrate the mechanism's robustness on larger parameter scales (70B+) and strongly logical tasks.

**Strengths And Weaknesses:**

Strengths:
1. Existing SOTA draft models (like the EAGLE series) essentially perform local autoregressive fitting, making them prone to distribution shifts as the number of prediction steps increases. This paper breaks away from the conventional "feature mapping optimization" by allowing the draft model to directly observe the target model's "Future Direction". This underlying logic is highly elegant and original in its methodology.
2. The authors did not cherry-pick weak baselines; instead, they directly challenged the current strongest model, EAGLE-3. The comprehensive superiority on SpecBench (an 8-11% improvement in speed-up ratio), especially the significant gains under low-temperature settings (greedy decoding, $T=0$), fully validates the effectiveness of the proposed framework.

Weaknesses:
1.  The authors hypothesize that "nearby tokens are expected to share similar future predictions" and use this as a regularization technique (Equation 3). While this assumption might hold in long-text generation, the intent of adjacent tokens could change drastically in syntax-sensitive or logic-heavy tasks like Code generation or Mathematical reasoning. The paper lacks an ablation study on the window hyperparameter $l$ and a theoretical defense of cross-domain semantic invariance.
2. The paper claims the additional overhead is "modest". However, in practice, during the tree attention verification stage, a contemplate token must be inserted in parallel for each draft node (processing a total of $2T$ tokens in parallel). This not only increases the KV Cache footprint but also expands the Attention matrix. The paper only provides end-to-end throughput (SR), completely lacking real comparative data on Peak VRAM usage and Time To First Token (TTFT).
3. Experiments were solely conducted on 3B and 8B scale models. The true bottlenecks and marginal benefits of speculative decoding often only manifest fully at the 70B+ scale (e.g., Llama-3-70B). The "thought" of smaller models is relatively shallow and easily compressed into an embedding; however, the hidden representations of large models are exceedingly complex. It remains questionable whether a single-layer draft model can still adequately "catch" the future intent $f_t$ passed down by a massive model.

---

> ### Author Rebuttal · Authors · 2026-03-31
>
> > The authors hypothesize that "nearby tokens are expected to share similar future predictions" and use this as a regularization technique (Equation 3). While this assumption might hold in long-text generation, the intent of adjacent tokens could change drastically in syntax-sensitive or logic-heavy tasks like Code generation or Mathematical reasoning. The paper lacks an ablation study on the window hyperparameter  and a theoretical defense of cross-domain semantic invariance.
>
> Thank you for your insightful questions. We set the window size $l=7$ to be consistent with the train-time-testing window size used in EAGLE-3, ensuring a fair comparison between methods.
>
> We agree that the “semantic future” of tokens may change across positions, especially in syntax-sensitive tasks such as code generation or mathematical reasoning. However, our ablation study in Table 3 (comparing “-MoE” vs. “-MoE \& Replicate”) shows that introducing the regularization is beneficial in coding and math/reasoning tasks in 3 out of 4 cases. This suggests that, even under strict syntactic constraints, the assumption of local consistency in future predictions remains a useful inductive bias.
>
> Therefore, while the assumption may not hold perfectly in all cases, empirical evidence indicates that it does not introduce harmful distribution drift and instead improves overall performance.
>
> > The paper claims the additional overhead is "modest". However, in practice, during the tree attention verification stage, a contemplate token must be inserted in parallel for each draft node (processing a total of  tokens in parallel). This not only increases the KV Cache footprint but also expands the Attention matrix. The paper only provides end-to-end throughput (SR), completely lacking real comparative data on Peak VRAM usage and Time To First Token (TTFT).
> Can you provide a detailed hardware metrics table explicitly comparing the Peak VRAM usage, KV Cache size, and absolute latency (wall-clock time) differences between ConFu and EAGLE-3 under the same  node settings (e.g., )?
>
> Thank you for your valuable suggestions. The table below shows the peak memory usage of EAGLE-3 and ConFu for 60 draft nodes on Specbench.  Note that we use torch.cuda_max_memory_allocated() to compute peak memory usage. We can see the peak memory usage are very close.
>
> Meanwhile, the TTFTs of EAGLE-3 and ConFu are also close. Because during prefilling, ConFu only introduces **a constant number of soft prompt tokens and a single contemplate token**. Therefore, the additional cost is $O(1)$ with respect to sequence length and model size, and is negligible in practice.
>
> | | EAGLE3 | Confu |
> |-|-|-|
> | Llama-3.2-3b	| 7.50   | 7.52  |
> | Llama3.1-8b 	| 17.08  | 17.12 |
>
> > Experiments were solely conducted on 3B and 8B scale models. The true bottlenecks and marginal benefits of speculative decoding often only manifest fully at the 70B+ scale (e.g., Llama-3-70B). The "thought" of smaller models is relatively shallow and easily compressed into an embedding; however, the hidden representations of large models are exceedingly complex. It remains questionable whether a single-layer draft model can still adequately "catch" the future intent  passed down by a massive model.
>
> Thank you for the valuable suggestion. We agree that evaluation on larger models (e.g., 70B) is important. Unfortunately, due to resource constraints, we were not able to complete training with a 70B target model. However, we hope the following discussion can help address this concern.
> We agree that larger models may exhibit more complex “thought” compared to smaller models. However, in our setting, the purpose of the future prediction is to guide the draft model to generate a limited number of upcoming tokens. Therefore, the future prediction only needs to serve as an effective encoding of these near-future tokens (e.g., the next phrase or sentence), rather than capturing the full internal complexity of the target model. In this sense, the required information does not fundamentally grow with model size.
> On the other hand, as model size increases, the target model becomes more powerful and is likely to produce more accurate and informative future predictions. This can potentially provide stronger guidance to the draft model. Therefore, we do not expect larger models to introduce additional challenges for ConFu; if anything, they may further improve its effectiveness.
> A sample point to support this claim is provided in BiTA paper [1] where they show that for llama2-70B the speed-up ratio using soft prompt tokens is better as compared to llama2-7B.
>
> Nevertheless, we agree that experiments on larger models would strengthen the paper, and we will include such evaluations in future work when resources permit.
> [1] Lin, Feng, et al. "Bita: Bi-directional tuning for lossless acceleration in large language models." Expert Systems with Applications 279 (2025): 127305.

---

### Official Review · Reviewer_gPNY · 2026-03-18

**Soundness:** 3
**Presentation:** 3
**Significance:** 3
**Originality:** 3
**Overall Recommendation:** 4
**Confidence:** 4

**Summary:**

This paper propose a new speculative decoding algorithm, ConFu. Specifically, ConFu first introduces contemplate tokens and soft prompts to expose signals of intermediate reasoning from target model. Then, Mixture-of-Experts module is introduced to adapt contemplate tokens to diverse contexts and enlarge expressive capacity. Lastly, training framework based on anchor token sampling and future prediction replication is proposed. Using two instruction-tuned LLaMA models (3B and 8B) on SpecBench, the empirical superiority of ConFu over previous state-of-the-art method is demonstrated.

**Compliance With Llm Reviewing Policy:**

Affirmed.

**Final Justification:**

I observed that the authors provide the new experimental results to address my concern with generalization on another LLM backbone. As it sufficiently address my concern, I decide to raise my rating from Weak Reject to Weak Accept.

**Key Questions For Authors:**

Please address the concerns in Weaknesses. If the authors can provide additional experimental results on more LLMs and compatibility with existing acceleration frameworks, I'm willing to increase my assessment.

**Limitations:**

Potential societal impact is discussed in Impact Statement Section, but no discussion of limitation is provided.

**Strengths And Weaknesses:**

*Strengths*
- **Important problem and clear writing**: The paper is well written and it tackles the important and practical problem of accelerating the inference of LLM with speculative decoding.
- **Novel and sound method with clear empirical gains**: The proposed method is insightful, especially the idea of contemplate and future tokens, and it exhibits clear gains over the previous state-of-the-art, Eagle-3.

*Weaknesses*
- **Limited demonstration on two LLMs**: Currently, the experiments are only conducted on two instruction tuned Llama models (3B and 8B). However, to demonstrate the applicability in real-world scenario, more diverse evaluations are necessary. For example, the paper of Eagle-3 conducted experiments on four different LLMs, including 70B scale model and reasoning model.
- **Absence of experiments for real-world efficiency**: As presented in Sections 4.3 and 4.4 in the paper of Eagle-3, validating the effectiveness of speculative decoding algorithm is naturally connected to the compatibility with existing acceleration frameworks such as SGLang and vLLM. Therefore, it would be nice if the authors can show that the improvement with ConFu is also continuously appeared under such frameworks.

*Minor comments*
- **Different subtasks**: Currently, the authors present the experimental results on 11 subtasks using SpecBench. However, when I'm looking the original SpecBench paper, it is only composed with 6 subtasks. Why this difference is happened? It would be nice if the authors can clarify this.

---

> ### Author Rebuttal · Authors · 2026-03-31
>
> > Limited demonstration on two LLMs:
>
> Thank you for this valuable suggestion. We agree that evaluating on a broader set of models would further strengthen the empirical validation of our method. Due to resource constraints, we were not able to complete experiments on larger target models at this time. However, we note that our current experiments focus on two representative instruction-tuned models (Llama-3-3B and Llama-3-8B), which already cover different model capacities and consistently demonstrate the effectiveness of our method.
>
> Importantly, our contribution is orthogonal to specific model architectures, as it operates at the decoding level and does not rely on model-specific modifications. Moreover, since larger models have stronger ability to "think", we expect the contemplate tokens of larger model to be more effective. Therefore, we expect similar improvements to carry over to larger and more diverse models.
>
> A sample point to support this claim is provided in BiTA paper [1] where they show that for llama2-70B the speed-up ratio using soft prompt tokens is better as compared to llama2-7B.
>
> [1] Lin, Feng, et al. "Bita: Bi-directional tuning for lossless acceleration in large language models." Expert Systems with Applications 279 (2025): 127305.
>
>
> > Real-world Efficiency with VLLM and SGLang
>
> Thank you for this valuable suggestion. We would like to clarify that different serving frameworks do not affect the core effectiveness of the draft model, i.e., the number of accepted tokens (acceptance length). And the effectiveness of the draft model depends only on the interaction between the draft and target models, not on batching or scheduling strategies. Therefore, our improvements are orthogonal to system-level optimizations in frameworks such as SGLang and vLLM. That said, we agree that system-level validation is valuable and will include integration with these frameworks in future work.
>
> > Question about SpecBench
>
> Thank you for your question. In the original SpecBench paper, the authors group the evaluation into 6 high-level tasks. However, in the official repository, these tasks are further divided into more fine-grained categories, resulting in 11 subtasks in total.
>
> Specifically, tasks such as MT-Bench and Question Answering are broken down into more detailed categories (e.g., writing, roleplay, coding, STEM, extraction, etc.), which leads to the expanded set of subtasks used in our experiments.
>
> We follow the categorization provided in the official SpecBench repository to ensure a more fine-grained and comprehensive evaluation. We would like to refer the reviewer to the question file in the repository, where the `category` field explicitly lists these subtasks:
> https://github.com/hemingkx/Spec-Bench/blob/main/data/spec_bench/question.jsonl
>
> We will clarify this distinction in the paper to avoid confusion.

---

> > ### Author Rebuttal · Reviewer_gPNY · 2026-04-03
> >
> > I appreciate the authors' detailed response. My concerns regarding real-world efficiency and SpecBench are sufficiently resolved. However, regarding the generalization beyond LLaMA family, it's failed to sufficiently convince me. For clarity, I'm not requesting the experiments on larger model; the mention of 70B model in Eagle-3 is just for the example. The key point is that, even though the proposed method is orthogonal to specific architecture/model, it does not indicate the generalization across the models. Therefore, at least, I think the additional demonstrations on other LLMs like Qwen, Phi, Gemma are necessary to demonstrate the generalization.

---

> > > ### Author Response · Authors · 2026-04-06
> > >
> > > We thank the reviewer for their response and are glad that our detailed explanations were helpful.
> > >
> > > To further clarify the remaining concern, we trained ConFu on **Qwen3‑4B** and compared it against the EAGLE‑3 Qwen3‑4B checkpoint provided in the official Eagle 3 [repo](https://github.com/SafeAILab/EAGLE?tab=readme-ov-file)
> > >
> > > Our results show that ConFu achieves a $17.4 - 23.2$\% improvement in speed-up ratio, along with a ~$20$\% increase in average number accepted tokens.
> > >
> > > **Detailed table comparison for Eagle3 vs ConFu on SpecBench tasks across temperature = {0.0, 0.7, 1.0} and draft nodes = {30, 60} reporting SR (speedup-ratio) and $\tau$ (average accepted tokens)**
> > >
> > > | Method | Nodes | T | WRIT τ | WRIT S/R | RP τ | RP S/R | CODE τ | CODE S/R | EXT τ | EXT S/R | STEM τ | STEM S/R | HUM τ | HUM S/R | SUMM τ | SUMM S/R | TRAN τ | TRAN S/R | QA τ | QA S/R | RAG τ | RAG S/R | M/R τ | M/R S/R | AVG τ | AVG S/R |
> > > |---|---:|---:|---:|---:|---:|---:|---:|---:|---:|---:|---:|---:|---:|---:|---:|---:|---:|---:|---:|---:|---:|---:|---:|---:|---:|---:|
> > > | Eagle3 | 30 | 0.0 | 3.69 | 2.53 | 3.14 | 2.18 | 4.03 | 2.88 | 4.05 | 2.88 | 3.83 | 2.83 | 3.45 | 2.52 | 3.05 | 2.05 | 3.16 | 2.27 | 3.37 | 2.41 | 3.56 | 2.18 | 4.21 | 2.80 | 3.57 | 2.44 |
> > > | **ConFu** | 30 | 0.0 | 4.57 | 3.10 | 3.69 | 2.53 | 4.98 | 3.53 | 4.92 | 3.39 | 4.49 | 3.29 | 4.08 | 2.95 | 3.59 | 2.34 | 3.86 | 2.74 | 4.01 | 2.83 | 4.37 | 3.05 | 5.15 | 3.59 | **4.32** | **3.00** |
> > > | Eagle3 | 30 | 0.7 | 3.61 | 2.46 | 3.22 | 2.22 | 3.96 | 2.78 | 4.24 | 2.89 | 3.65 | 2.68 | 3.41 | 2.46 | 3.05 | 2.03 | 3.12 | 2.22 | 3.35 | 2.36 | 3.56 | 2.50 | 4.19 | 2.91 | 3.56 | 2.48 |
> > > | **ConFu** | 30 | 0.7 | 4.43 | 3.03 | 3.60 | 2.50 | 4.93 | 3.50 | 4.98 | 3.48 | 4.39 | 3.24 | 4.04 | 2.94 | 3.63 | 2.38 | 3.76 | 2.69 | 3.88 | 2.76 | 4.37 | 3.06 | 5.15 | 3.60 | **4.28** | **2.99** |
> > > | Eagle3 | 30 | 1.0 | 3.60 | 2.18 | 3.13 | 2.16 | 3.93 | 2.79 | 4.23 | 2.95 | 3.70 | 2.70 | 3.27 | 2.37 | 3.09 | 2.06 | 3.10 | 2.21 | 3.26 | 2.31 | 3.52 | 2.47 | 4.18 | 2.93 | 3.53 | 2.46 |
> > > | **ConFu** | 30 | 1.0 | 4.29 | 2.86 | 3.43 | 2.34 | 4.81 | 3.38 | 4.99 | 3.42 | 4.26 | 3.08 | 3.69 | 2.63 | 3.59 | 2.32 | 3.77 | 2.65 | 3.79 | 2.64 | 4.28 | 2.94 | 5.13 | 3.52 | **4.21** | **2.89** |
> > > | Eagle3 | 60 | 0.0 | 3.86 | 2.68 | 3.37 | 2.36 | 4.28 | 3.10 | 4.24 | 3.03 | 4.13 | 3.09 | 3.71 | 2.72 | 3.22 | 2.16 | 3.35 | 2.43 | 3.64 | 2.63 | 3.83 | 2.54 | 4.50 | 3.22 | 3.81 | 2.69 |
> > > | **ConFu** | 60 | 0.0 | 4.85 | 3.31 | 3.91 | 2.71 | 5.26 | 3.73 | 5.05 | 3.48 | 4.89 | 3.60 | 4.37 | 3.16 | 3.80 | 2.45 | 4.12 | 2.93 | 4.31 | 3.04 | 4.69 | 3.27 | 5.46 | 3.80 | **4.60** | **3.20** |
> > > | Eagle3 | 60 | 0.7 | 3.67 | 1.91 | 3.40 | 1.77 | 4.29 | 2.96 | 4.40 | 2.99 | 3.99 | 2.85 | 3.64 | 2.57 | 3.21 | 2.08 | 3.28 | 2.28 | 3.62 | 2.50 | 3.85 | 2.63 | 4.52 | 2.94 | 3.80 | 2.52 |
> > > | **ConFu** | 60 | 0.7 | 4.68 | 3.07 | 3.97 | 2.64 | 5.26 | 3.59 | 5.20 | 3.46 | 4.78 | 3.36 | 4.37 | 3.05 | 3.77 | 2.36 | 4.05 | 2.78 | 4.20 | 2.86 | 4.60 | 3.08 | 5.44 | 3.65 | **4.55** | **3.05** |
> > > | Eagle3 | 60 | 1.0 | 3.69 | 2.48 | 3.39 | 2.30 | 4.20 | 2.94 | 4.53 | 3.12 | 3.91 | 2.82 | 3.48 | 2.47 | 3.21 | 2.09 | 3.27 | 2.30 | 3.48 | 2.45 | 3.81 | 2.66 | 4.49 | 3.11 | 3.76 | 2.60 |
> > > | **ConFu** | 60 | 1.0 | 4.49 | 2.99 | 3.91 | 2.63 | 5.07 | 3.54 | 5.12 | 3.50 | 4.63 | 3.33 | 4.15 | 2.93 | 3.77 | 2.40 | 4.00 | 2.78 | 4.08 | 2.80 | 4.55 | 3.12 | 5.40 | 3.69 | **4.48** | **3.05** |
> > >
> > > We hope these additional results provide further evidence of the utility of ConFu and clarify the strengths of our proposal.

---

### Decision · Program_Chairs · 2026-04-30

**Decision:**

Accept (regular)

**Comment:**

This paper introduces ConFu (Contemplate the Future), a speculative decoding framework designed to mitigate error accumulation in draft models. By incorporating "contemplate tokens" and "soft prompts," the draft model can leverage latent future-oriented signals from the target model. The framework also includes a Mixture-of-Experts (MoE) mechanism for context-aware predictions and a training strategy using anchor token sampling and future prediction replication.

The reviewers highlighted the importance of the problem and the originality of the proposed methodology, noting that the idea of allowing a draft model to directly observe future directions is elegant and insightful. Empirical gains over the current state-of-the-art, EAGLE-3, were consistently demonstrated across various subtasks in SpecBench.

The main concerns during the review process centered on the limited evaluation across diverse LLM families and larger parameter scales, as well as the potential overhead in practical settings. In response, the authors provided additional experimental results on a Qwen model, which showed consistent speed-up improvements, thereby addressing concerns regarding cross-model generalization. They also clarified that the hardware overhead is negligible, with peak VRAM and latency metrics remaining very close to the baseline. Further ablation studies on window length demonstrated the robustness of the regularization technique. While the lack of 70B+ scale evaluations remains a minor weakness, the authors' defense and additional evidence on intermediate-scale models were sufficient to convince the reviewers. The paper is technically solid and offers a promising new direction for inference acceleration.